# A Universal Music Translation Network

**Noam Mor**
Facebook AI Research
noam.mor@gmail.com

**Lior Wolf & Adam Polyak**
Facebook AI Research & Tel Aviv Uni.
wolf,adampolyak@fb.com

**Yaniv Taigman**
Facebook AI Research
yaniv@fb.com

## Abstract

We present a method for translating music across musical instruments and styles. This method is based on unsupervised training of a multi-domain wavenet autoencoder, with a shared encoder and a domain-independent latent space that is trained end-to-end on waveforms. Employing a diverse training dataset and large net capacity, the single encoder allows us to translate also from musical domains that were not seen during training. We evaluate our method on a dataset collected from professional musicians, and achieve convincing translations. We also study the properties of the obtained translation and demonstrate translating even from a whistle, potentially enabling the creation of instrumental music by untrained humans.

## 1 Introduction

Humans have always created music and replicated it – whether it is by singing, whistling, clapping, or, after some training, playing improvised or standard musical instruments. This ability is not unique to us, and there are many other vocal mimicking species that are able to repeat music from hearing. Music is also one of the first domains to be digitized and processed by modern computers and algorithms. It is, therefore, somewhat surprising that in the core music task of mimicry, AI is still much inferior to biological systems.

In this work, we present a novel way to produce convincing musical translation between instruments and styles. For example[1], we convert the audio of a Mozart symphony performed by an orchestra to an audio in the style of a pianist playing Beethoven. Our ability builds upon two technologies that have recently become available: (i) the ability to synthesize high quality audio using autoregressive models, and (ii) the recent advent of methods that transform between domains in an unsupervised way. The first technology allows us to generate high quality and realistic audio and thanks to the teacher forcing technique, autoregressive models are efficiently trained as decoders. The second family of technologies contributes to the practicality of the solution, since posing the learning problem in the supervised setting, would require a parallel dataset of different musical instruments.

In our architecture, we employ a single, universal, encoder and apply it to all inputs (universal here means that a single encoder can address all input music, allowing us to achieve capabilities that are known as universal translation). In addition to the advantage of training fewer networks, this also enables us to convert from musical domains that were not heard during training to any of the domains encountered.

The key to being able to train a single encoder architecture, is making sure that the domain-specific information is not encoded. We do this using a domain confusion network that provides an adversarial signal to the encoder. In addition, it is important for the encoder not to memorize the input signal but to encode it in a semantic way. We achieve this by distorting the input audio by random local pitch modulation. During training, the network is trained as a denoising autoencoder, which recovers the undistorted version of the original input. Since the distorted input is no longer in the musical domain of the output, the network learns to project out-of-domain inputs to the desired output domain. In addition, the network no longer benefits from memorizing the input signal and employs a higher-level encoding.

---

[1]Audio samples are available anonymously as supplementary at: musictranslation.github.io/

Asked to convert one musical instrument to another, our network shows a level of performance that seems to approach that of musicians. When controlling for audio quality, which is still lower for generated music, it is many times hard to tell which is the original audio file and which is the output of the conversion that mimics a completely different instrument. The network is also able to successfully process unseen musical instruments such as drums, or other sources, such as whistles.

## 2 PREVIOUS WORK

**Domain Transfer** Recently, there has been a considerable amount of work, mostly on images and text, which performs unsupervised translation between domains $\mathcal{A}$ and $\mathcal{B}$, without being shown any matching pairs, i.e., in a completely unsupervised way. Almost all of this work employs GAN constraints (Goodfellow et al., 2014), in order to ensure a high level of indistinguishability between the translations of samples in $A$ and samples from the domain $B$. In our work, the output is generated by an autoregressive model and training takes place using the ground truth output of the previous time steps ("teacher forcing"), instead of the predicted ones. A complete autoregressive inference is only done during test time, and it is not practical to apply such inference during training in order to get a realistic generated ("fake") sample for the purpose of training the GAN.

Another popular constraint is that of circularity, namely that by mapping from $\mathcal{A}$ to $\mathcal{B}$ and back to $\mathcal{A}$ a reconstruction of the original sample is obtained (Kim et al., 2017; Zhu et al., 2017; Yi et al., 2017). In our work, for the same reason mentioned above, the output during training does not represent the future test time output, and such a constraint is unrealistic. An application of circularity in audio was present in (Kaneko & Kameoka, 2017), where a non-autoregressive model between vocoder features is used to convert between voices in an unsupervised way.

Cross domain translation is not restricted to a single pair of domains. The recent StarGAN (Choi et al., 2017) method creates multiple cycles for mapping between multiple (more than two) domains. The method employs a single generator that receives as input the source image as well as the specification of the target domain. It then produces the analog "fake" image from the target domain. Our work employs multiple decoders, one per domain, and attempts to condition a single decoder on the selection of the output domain failed to produce convincing results.

UNIT (Liu et al., 2017) employs an encoder-decoder pair per each domain, where the latent spaces of the domains are assumed to be shared. This is achieved by sharing the network layers that are distant from the image (the top layers of the encoder and the bottom layers of the decoder), similarly to CoGAN (Liu & Tuzel, 2016). Cycle-consistency is also added, and structure is added to the latent space using a variational autoencoder (Kingma & Welling, 2014) loss terms. Our method employs a single encoder, which eliminates the need for many of the associated constraints. In addition, we do not impose a VAE loss term (Kingma & Welling, 2014) on the latent space of the encodings and instead employ a domain confusion loss (Ganin et al., 2016). The work of Louizos et al. (2015) investigates the problem of learning invariant representations by employing the Maximum Mean Discrepancy (MMD), which we do not use.

**Audio Synthesis** WaveNet (van den Oord et al., 2016) is an autoregressive model that predicts the probability distribution of the next sample, given the previous samples and an input conditioning signal. Its generated output is currently considered of the highest naturalness, and is applied in a range of tasks. In (Rethage et al., 2017), the authors have used it for denoising waveforms by predicting the middle ground-truth sample from its noisy input support. Recent contributions in Text-To-Speech(TTS) (Ping et al., 2018; Shen et al., 2018) have successfully conditioned wavenet on linguistic and acoustic features to obtain state of the art performance.

In VQ-VAE (van den Oord et al., 2017), voice conversion was obtained by employing a variational autoencoder that produces a quantized latent space that is conditioned on the speaker identity. Similar to our work, the decoder is based on WaveNet. However, we impose a greater constraint on the latent space by (a) having a universal encoder, forcing the embeddings of all domains to lie in the same space, yet (b) training a separate reconstructing decoder for each domain, provided that (c) the latent space is domain independent, thereby reducing source-target pathways memorization, which is also accomplished by (d) employing augmentation to distort the input signal. Invariance is achieved in VQ-VAE through the strong bottleneck effect achieved by discretization. Despite some effort, we were not able to use a discrete latent space here.

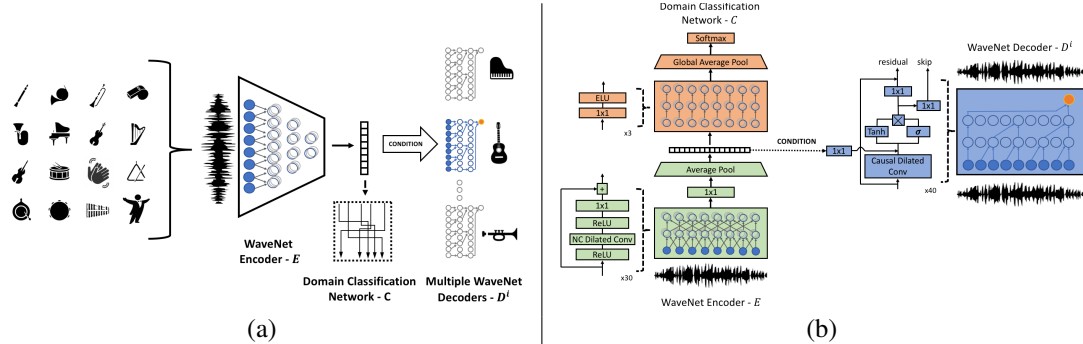

Figure 1: (a) The schematic architecture of our translation network. The confusion term (marked by the dashed line) is employed only during training. $E$ is the shared encoder, $C$ is the domain classification network employed in the domain confusion term, $D^i$ are the various decoders. (b) A detailed depiction of our architecture. 'NC' indicates non-causal convolution. '1x1' indicates a 1-D convolution with kernel size 1.

Recently, Dieleman et al. (2018) explored discretization as a method to capture long-range dependencies in unconditioned music generation, for up to 24 seconds. We focus on translation, and the conditioning on the source signal carries some long-range information on the development of the music. Consider an analogy to a myopic language translation system, where the input is a story in English and the output is a story in Spanish. Even if the translation occurs one sentence at a time, the main theme of the story is carried by the "conditioning" on the source text.

The architecture of the autoencoder we employ is the wavenet-autoencoder presented in (Engel et al., 2017). In comparison to this work, our inputs are not controlled and are collected from consumer media. Our overall architecture differs in that multiple decoders and an additional auxiliary network, which is used for disentangling the domain information from the other aspects of the music representation, are trained and by the addition of an important augmentation step.

In the supervised learning domain, an audio style transfer between source and target spectrograms was performed with sequence-to-sequence recurrent networks (Haque et al., 2018). This method requires matching pairs of samples played on different instruments. In another fully supervised work (Hadjeres & Pachet, 2017), a graphical model aimed at modeling polyphonic tones of Bach was trained on notes, capturing the specificity of Bach's chorales. This model is based on RNNs and requires a large corpus of notes of a particular instrument produced with a music editor.

**Style Transfer** Style transfer is often confused with domain translation and the distinction is not always clear. In the task of style transfer, the "content" remains the same between the input and the output, but the "style" is modified. Notable contributions in the field include (Gatys et al., 2016; Ulyanov et al., 2016; Johnson et al., 2016), which synthesize a new image that minimizes the content loss with respect to the content-donor sample and the style loss with respect to one or more samples of a certain style. The content loss is based on comparing the activations of a network training for an image categorization task. The style loss compares the statistics of the activations in various layers of the categorization layer. An attempt at audio style transfer is described in (Barry & Kim, 2018).

**Concatenative Synthesis** In the computer music and audio effects literature, the conversion task we aim to solve is tackled by concatenating together short pieces of audio from the target domain, such that the output audio resembles the input audio from the source domain Verfaille & Arfib (2000); Schwarz (2006); Zils & Pachet (2001); Simon et al. (2005). The method has been extensively researched, see the previous work section of Nuanáin et al. (2017) and the online resource of Schwarz (2018). A direct comparison to such methods is challenging, since many of the methods have elaborate interfaces with many tunable parameters that vary from one conversion task to the next. To the extent possible, we compare with some of the published results in Sec. 4.2, obtaining what we believe to be clearly superior results.

## 3 METHOD

Our domain translation method is based on training multiple autoencoder pathways, one per musical domain, such that the encoders are shared. During training, a softmax-based reconstruction loss is applied to each domain separately. The input data is randomly augmented, prior to applying the encoder, in order to force the network to extract high-level semantic features, instead of simply memorizing the data. In addition, a domain confusion loss (Ganin et al., 2016) is applied to the latent space to ensure that the encoding is not domain-specific. A diagram of the translation architecture is shown in Fig. 1 (a).

### 3.1 WAVENET AUTOENCODER

We reuse an existing autoencoder architecture that is based on a WaveNet decoder and a WaveNet-like dilated convolution encoder (Engel et al., 2017). The WaveNet of each decoder is conditioned on the latent representation produced by the encoder. In order to reduce the inference-time, the nv-wavenet CUDA kernels provided by NVIDIA ( `https://github.com/NVIDIA/nv-wavenet`) were used after modification to better match the architecture suggested by van den Oord et al. (2016), as described below.

The encoder is a fully convolutional network that can be applied to any sequence length. The network has three blocks of ten residual-layers, a total of thirty layers. Each residual-layer contains a RELU nonlinearity, a non-causal dilated convolution with an increasing kernel size, a second RELU, and a $1 \times 1$ convolution followed by the residual summation of the activations before the first RELU. There is a fixed width of 128 channels. After the three blocks, there is an additional $1 \times 1$ layer. An average pooling with a kernel size of 50 milliseconds (800 samples) follows in order to obtain an encoding in $\mathbb{R}^{64}$, which implies a temporal down sampling by a factor of $\times 12.5$.

The encoding is upsampled temporally to the original audio rate, using nearest neighbor interpolation and is used to condition a WaveNet decoder. The conditioning signal is passed through a $1 \times 1$ layer that is different for each WaveNet layer. The audio (both input and output) is quantized using 8-bit mu-law encoding, similarly to both (van den Oord et al., 2016; Engel et al., 2017), which results in some inherent loss of quality. The WaveNet decoder has either four blocks of 10 residual-layers and a resulting receptive field of 250 milliseconds (4,093 samples), as in Engel et al. (2017), or 14 layer blocks and a much larger receptive field of 4 seconds. Each residual-layer contains a causal dilated convolution with an increasing kernel size, a gated hyperbolic tangent activation, a $1 \times 1$ convolution followed by the residual summation of the layer input, and a $1 \times 1$ convolution layer which introduces a skip connection. Each residual-layer is conditioned on the encoding described above. The summed skip connections are passed through two fully connected layers and a softmax activation to output the next timestep probability. A detailed diagram of the WaveNet autoencoder is shown in Fig. 1 (b).

We modify the fast nv-wavenet CUDA inference kernels, which implement the architecture suggested by Ping et al. (2018), and create efficient WaveNet kernels that implement the WaveNet architecture suggested by Engel et al. (2017). Specifically, we make the following modifications to nv-wavenet: (i) we add initialization of skip connections with previous WAV samples, (ii) we increase the kernel capacity to support 128 residual channels and (iii) we also add the conditioning to the last fully connected layer.

### 3.2 AUDIO INPUT AUGMENTATION

In order to improve the generalization capability of the encoder, as well as to enforce it to maintain higher-level information, we employ a dedicated augmentation procedure that changes the pitch locally. The resulting audio is of a similar quality but is slightly out of tune.

Specifically, we perform our training on segments of one second length. For augmentation, we uniformly select a segment of length between 0.25 and 0.5 seconds, and modulate its pitch by a random number between -0.5 and 0.5 of half-steps, using librosa (McFee et al., 2015).

### 3.3 TRAINING AND THE LOSSES USED

Let $s^j$ be an input sample from domain $j = 1, 2, \ldots, k$, $k$ being the number of domains employed during training. Let $E$ be the shared encoder, and $D^j$ the WaveNet decoder for domain $j$. Let $C$ be the domain classification network, and $O(s, r)$ be the random augmentation procedure applied to a sample $s$ with a random seed $r$.

The network $C$ predicts which domain the input data came from, based on the latent vectors. It applies three 1D-convolution layers, with the ELU (Clevert et al., 2017) nonlinearity. The last layer projects the vectors to dimension $k$ and the vectors are subsequently averaged to a single $\mathbb{R}^k$ vector. A detailed diagram of network $C$ is shown as part of Fig. 1 (b).

During training, the domain classification network $C$ minimizes the classification loss

$$\Omega = \sum_j \sum_{s^j} \mathbb{E}_r \, \mathcal{L}(C(E(O(s^j, r))), j), \tag{1}$$

and the music to music autoencoders $j = 1, 2, \ldots$ are trained with the loss

$$-\lambda\Omega + \sum_j \sum_{s^j} \mathbb{E}_r \mathcal{L}(D^j(E(O(s^j, r))), s^j) \tag{2}$$

where $\mathcal{L}(o, y)$ is the cross entropy loss applied to each element of the output $o$ and the corresponding element of the target $y$ separately. Note that the decoder $D^j$ is an autoregressive model that is conditioned on the output of $E$. During training, the autoregressive model is fed the target output $s^j$ from the previous time-step, instead of the generated output.

### 3.4 NETWORK DURING INFERENCE

To perform the actual transformation from a sample $s$ from any domain, even from an unseen musical domain, to output domain $j$, we apply the autoencoder of domain $j$ to it, without applying the distortion. The new sample $\hat{s}^j$ is, therefore, given as $D^j(E(s))$. The bottleneck during inference is the WaveNet autoregressive process, which is optimized by the dedicated CUDA kernels.

## 4 EXPERIMENTS

We conduct music translation experiments, using a mix of human evaluation and qualitative analysis, in order to overcome the challenges of evaluating generative models. The experiments were done in two phases. In the first phase, described in an earlier technical report (Mor et al., 2018), we train our network on six arbitrary classical musical domains: (i) Mozart's symphonies conducted by Karl Böhm, (ii) Haydn's string quartets, performed by the Amadeus Quartet, (iii) J.S Bach's cantatas for orchestra, chorus and soloists, (iv) J.S Bach's organ works, (v) Beethoven's piano sonatas, performed by Daniel Barenboim, and (vi) J.S Bach's keyboard works, played on Harpsichord. The music recordings by Bach (iii,iv,vi) are from the Teldec 2000 Complete Bach collection.

In the second phase, in order to allow reproducibility and sharing of the code and models, we train on audio data from MusicNET (Thickstun et al., 2017). Domains were chosen as the largest domains that show variability between composers and instruments. The following six domains were selected: (i) J.S Bach's suites for cello, (ii) Beethoven's piano sonatas, (iii) Cambini's Wind Quintet, (iv) J.S Bach's fugues, played on piano, (v) Beethoven's violin sonatas and (vi) Beethoven's string quartet. This public dataset is somewhat smaller than the data used in phase one.

The phases differ in the depth of the decoders: in the first phase, we employed blocks of ten layers, while in the second, we shifted to larger receptive fields and blocks of 14. The training and test splits are strictly separated by dividing the tracks (or audio files) between the two sets. The segments used in the evaluation experiments below were not seen during training. During training, we iterate over the training domains, such that each training batch contains 16 randomly sampled one second samples from a single domain. Each batch is first used to train the domain classification network $C$, and then to train the universal encoder and the domain decoder, given the updated discriminator.

The method was implemented in the PyTorch framework, and trained on eight Tesla V100 GPUs for a total of 6 days. We used the ADAM optimization algorithm with a learning rate of $10^{-3}$ and a decay factor of $0.98$ every 10,000 samples. We weighted the confusion loss with $\lambda = 10^{-2}$.

Table 1: MOS scores (mean± SD) for the conversion tasks.

| Converter | Harpsichord→ Piano | | Orchestra→ Piano | | New domains→ Piano | |
|---|---|---|---|---|---|---|
| | Audio quality | Translation success | Audio quality | Translation success | Audio quality | Translation success |
| Musician E | 3.89 ± 1.06 | 4.10± 0.94 | 4.02± 0.81 | 4.12± 0.97 | 4.44±0.82 | 4.13± 0.83 |
| Musician M | 3.82 ± 1.18 | 3.75± 1.17 | 4.13± 0.89 | 4.12± 0.98 | 4.48±0.72 | 3.97± 0.88 |
| Musician A | 3.69 ± 1.08 | 3.91± 1.16 | 4.06± 0.86 | 3.99± 1.08 | 4.53±0.79 | 3.93± 0.95 |
| Our | 2.95 ± 1.18 | 3.07± 1.30 | 2.56± 1.04 | 2.86± 1.16 | 2.36±1.17 | 3.18± 1.14 |

## 4.1 EVALUATION OF TRANSLATION QUALITY

The first set of experiments compared the method to human musicians using the phase one network. Since human musicians, are equipped by evolution with music skills, selected among their peers according to their talent, and who have trained for decades, we do not expect to do better than humans at this point. To perform this comparsion, music from domain $X$ was converted to piano, for various $X$. The piano was selected for practical reasons: pianists are in higher availability than other musicians and a piano is easier to produce than, e.g., an orchestra.

Three professional musicians with a diverse background were employed for the conversion task: E, who is a conservatory graduate with an extensive background in music theory and piano performance, and also specializes in transcribing music; M, who is a professional producer, composer, pianist and audio engineer, who is an expert in musical transcription; and A who is a music producer, editor, and a skilled player of keyboards and other instruments.

The task used for comparison was to convert 60 segments of five seconds each to piano. Three varied sources were used. 20 of the segments were from Bach's keyboard works, played on a Harpsichord, and 20 others were from Mozart's 46 symphonies conducted by Karl Böhm, which are orchestral works. The last group of 20 segments was a mix of three different domains that were not encountered during training – Swing Jazz, metal guitar riffs, and instrumental Chinese music. The 60 music segments were encoded by the universal encoder and decoded by the WaveNet trained on Beethoven's piano sonatas, as performed by Daniel Barenboim.

In order to compare between the conversions, we employed human evaluation, which is subjective and could be a mix of the assessment of the audio quality and the assessment of the translation itself. This limits the success of the automatic method, since the quality of the algorithm's output is upper bounded by the neural network architecture and cannot match that of a high quality recording.

Since there is a trade-off between the fidelity to the original piece and the ability to create audio in the target domain, we present two scores: audio quality of the output piano and a matching score for the translation. While one can argue that style is hard to define and, therefore, such subjective experiments are not well founded, there are many similar MOS experiments in image to image translation, e.g., (Lample et al., 2017), and indeed MOS studies are used exactly where the translation metric is perceptual and subjective.

Specifically, Mean Opinion Scores (MOS) were collected using the CrowdMOS (Ribeiro et al., 2011) package. Two questions were asked: (1) what is the quality of the audio, and (2) how well does the converted version match the original. The results are shown in Tab. 1. It shows that our audio quality is considerably lower than the results produced by humans, using a keyboard connected to a computer (which should be rated as near perfect and makes any other audio quality in the MOS experiment pale in comparison). Regarding the translation success, the conversion from Harpsichord is better than the conversion from Orchestra. Surprisingly, the conversion from unseen domains is more successful than both these domains. In all three cases, our system is outperformed by the human musicians, whose conversions will soon be released to form a public benchmark.

**Lineup experiment** In another set of experiments, we evaluate the ability of persons to identify the source musical segment from the conversions. We present, in each test, a set of six segments. One segment is a real segment from a random domain out of the ones used to train our network, and five are the associated translations. We shuffle the segments and ask which is the original one and

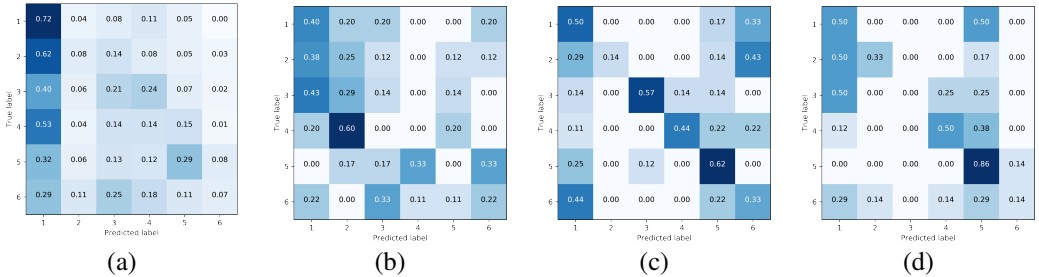

Figure 2: Results of the lineup experiment. (a) listeners from the general population tend to select the same domain as the source regardless of the actual source. (b) the musician A failed to identify the source most of the time. (c) the amateurs T and (d) S failed most of the time.

which are conversions. To equate the quality of the source to that of the translations and prevent identification by quality, we attach the source after passing it through its domain's autoencoder.

The translation is perfectly authentic, if the distribution of answers is uniform. However, the task is hard to define. In a first attempt, Amazon Mechanical Turk (AMT) freelancers tended to choose the Mozart domain as the source, regardless of the real source and the presentation order, probably due to its relatively complex nature in comparison to the other domains. This is shown in the confusion matrix of Fig. 2(a). We, therefore, asked two amateur musicians (T, a guitarist, and S a dancer and a drummer with a background in piano) and the professional musician A (from the first experiment) to identify the source sample out of the six options, based on authenticity.

The results, in Fig. 2(b-d) show that there is a great amount of confusion. T and A failed in most cases, and A tended to show a similar bias to the AMT freelancers. S also failed to identify the majority of the cases, but showed coherent confusion patterns between pairs of instruments.

**NSynth pitch experiments** NSynth (Engel et al., 2017) is an audio dataset containing samples of 1,006 instruments, each sample labeled with a unique pitch, timbre, and envelope. Each sample is a four second monophonic 16kHz snippet, ranging over every pitch of a standard MIDI piano (21-108) as well as five different velocities. It was not seen during training of our system.

We measure the correlation of embeddings retrieved using the encoder of our network across pitch for multiple instruments. The first two columns (from the left hand side) of Fig. 3 show self-correlations, while the third column shows correlation across instruments. As can be seen, the embedding encodes pitch information very clearly, despite being trained on complex polyphonic audio. The cosine similarity between the two instruments for the same pitch is, on average, 0.90-0.95 (mean of the diagonal), depending on the pair of instruments.

## 4.2 EXPLORATORY EXPERIMENTS

In order to freely share our trained models and allow for maximal reproducibility, we have retrained the network with data from MusicNet (Thickstun et al., 2017). The following experiments are based on this network and are focused on understanding the properties of the conversion. The description is based on the supplementary media available at `musictranslation.github.io`.

**Are we doing more than timbral transfer?** Is our system equivalent to pitch estimation followed by rendering with a different instrument, or can it capture stylistic musical elements? We demonstrate that our system does more than timbral transfer in two ways. Consider the conversions presented in supplementary S1, which consist of many conversion examples from each of the domains to every other domain. There are many samples where it is clear that more than timbral transfer is happening. For example, when converting Beethoven's string quartet music to a wind quintet (Sample #30), an ornamentation note is added in the output that is nowhere to be found in the input music; when converting Beethoven's violin sonata to Beethoven's solo piano (Samples #24 and #23), the violin line seamlessly integrated into the piano part; when converting Beethoven's solo piano music to Bach's solo cello (Sample #9), the bass line of the piano is converted to cello. It is perhaps most evident when converting solo piano to piano and violin; an identity transformation would have been a valid translation, but the network adds a violin part to better match the output distribution.

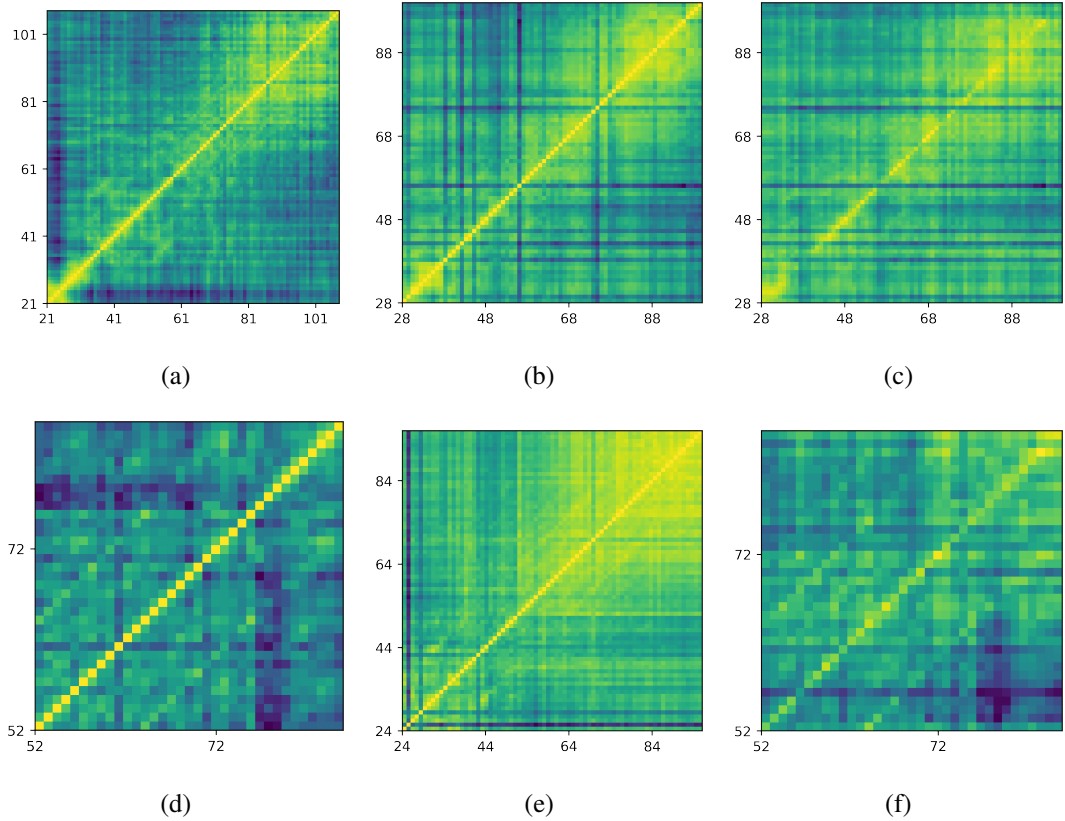

Figure 3: Correlation of embeddings across pitch. (a) Self-correlation for NSynth's flute-acoustic-027. (b) Self-correlation for keyboard-electronic-019. (c) The correlation between the electronic keyboard (y-axis) and the flute. (d) Self-correlation for brass-acoustic-018. (e) Self-correlation for string-acoustic-029. (f) The correlation between the brass instrument (y-axis) and the string.

To further demonstrate the capabilities of our system, we train a network on two piano domains: MusicNet solo piano recordings of Bach and Beethoven. We reduce the size of the latent space to 8 to limit the ability of the original input to be repeated exactly, thereby encouraging the decoders to be more "creative" than they normally would, with the goal of observing how decoders trained on different training data will use their freedom of expression. The input we employ is a simple MIDI synthesized as a piano. Supplementary S2 presents a clear stylistic difference: one can hear some counterpoint in the Bach sample, whereas the Beethoven output exhibits a more "Sturm und Drang" feeling, indicating that the network learns stylistic elements from the training data.

**Comparison with previous methods** We compare our results with those of Concatenative Synthesis methods in supplementary S3. To do that, we use our system to translate target files from published results of two works in that field, and present the methods' results side-by-side. Samples 1 and 2 are compared with the published results of Coleman (2016), a work comparing several Concatenative Synthesis methods, and uses a violin passage as source audio input. Sample 3 is compared with MATConcat (Sturm, 2006), which uses a corpus of string quartets as a source material. Sample 1 is a fugue performed on a piano. We show that we are able to convincingly produce string quartet and wind ensemble renditions of the piece. To push our model to its boundaries, we also attempt to convert the polyphonic fugue to solo cello, obtaining a rather convincing result. We believe that our results surpass in naturalness those obtained by concatenative methods. Sample 2 is an orchestra piece, which for our system is data that has never been seen during training. We convert it to piano, solo cello and a wind quintet, achieving convincing results, that we believe surpass the concatenative synthesis results. Sample 3 is another orchestra piece, which includes a long drum roll, followed by brass instruments. It is not quite well-defined how to convert a drum roll to a string quartet, but we believe our rendition is more coherent. Our method is able to render the brass instruments and

orchestral music after the drum roll more convincingly than MATConcat, which mimics the audio volume but loses most musical content.

**Universality** Note that our network has never observed drums, brass instruments or an entire orchestra during training, and, therefore, the results of supplementary S3 also serve to demonstrate the versatility of the encoder module resulting from our training procedure (and so do those of S2). Supplementary S4 presents more out-of-domain conversion results, including other domains from MusicNet, whistles, and even spontaneous hand clapping.

The universality property hinges on the success of training a domain-independent representation. As can be seen in the confusion matrices given in Fig. 4, the domain classification network does not do considerably better than chance when the networks converge.

**Ablation Analysis** We conducted three ablation studies. In the first study, the training procedure did not use the augmentation procedure of Sec. 3.2. This resulted in a learning divergence during training, and we were unable to obtain a working model trained without augmentation, despite considerable effort.

In order to investigate the option of not using augmentation in a domain where training without it converges, we have applied our method to the task of voice conversion. Our experiments show a clear advantage for applying augmentation, see Appendix A. Additional experiments were conducted, for voice conversion, using the VQ-VAE method of van den Oord et al. (2017).

In the second ablation study, the domain classification network was not used ($\lambda = 0$). Without requiring that the shared encoder remove domain-specific information from the latent representation of the input, the network learned to simply encode all information in the latent vectors, and all decoders learned to turn this information back to the original waveform. This resulted in a model that does not do any conversion at all.

Finally, we performed an ablation study on the latent code size, in which we convert a simple MIDI clip to the Beethoven domain and the Bach domain. Samples are available as supplementary S6. As can be heard, a latent dimensionality of 64 tends to reconstruct the input (unwanted memorization). A model with a latent space of 8 (used in S2) performs well. A model with a latent dimensionality of 4 is more creative, less related to the input midi, and also suffers from a reduction in quality.

**Semantic blending** We blend two encoded musical segments linearly in order to check the additivity of the embedding space. For that, we have selected two random five second segments $i$ and $j$ from each domain and embedded both using the encoder, obtaining $e_i$ and $e_j$. We then combine the embeddings as follows: starting with 3.5 seconds from $e_i$, we combine the next 1.5 seconds of $e_i$ with the first 1.5 seconds of $e_j$ using a linear weighting with weights $1 - t/1.5$ and $t/1.5$ respectively, where $t \in [0, 1.5]$. We then use the various decoders to generate audio. The results are natural and the shift is completely seamless, as far as we observe. See supplementary S5 for samples.

The samples also demonstrate that in the scenario we tested, one can alternatively use fade-in and fade-out to create a similar effect. We therefore employ a second network that is used for a related task of voice conversion (see Appendix A) and demonstrate that in the case of voice conversion, latent space embedding is clearly superior to converting the audio itself. These samples can also be found in supplementary S5 for details and samples.

## 5 DISCUSSION

Our work demonstrates capabilities in music conversion, which is a high-level task (a terminology that means that they are more semantic than low-level audio processing tasks), and could open the door to other high-level tasks, such as composition. We have initial results that we find interesting: by reducing the size of the latent space, the decoders become more "creative" and produce outputs that are natural yet novel, in the sense that the exact association with the original input is lost.

ACKNOWLEDGMENTS

This work is part of Adam Polyak's Ph.D thesis research conducted at Tel Aviv University.

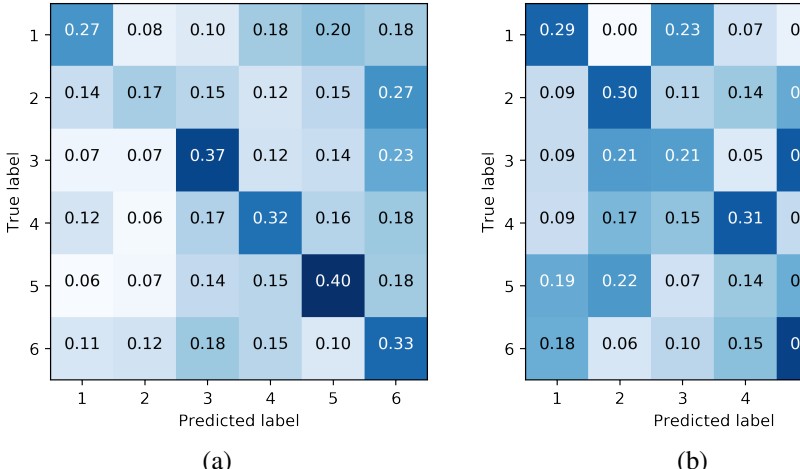

Figure 4: Accuracy of the domain classification network. (a) A confusion matrix of the domain classification network at the end of training on the private dataset used in the first phase of experiments. The mean accuracy is 0.30. (b) The confusion matrix for the MusicNet dataset used in the second phase of experiments. The mean accuracy is 0.24.

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

## A  VOICE CONVERSION EXPERIMENTS

We further evaluate our method on the task of voice conversion, which is not as challenging as the music conversion task explored in this work. It is, therefore, a convenient test bed when comparing to the VQ-VAE (van den Oord et al., 2017) method, which, as we mention in the paper, did not perform well in our music-based experiments, and which was shown by the authors to work on voice conversion.

In addition, as mentioned in Sec. 4.2, successful training on the music domains requires data augmentation. In voice conversion, we were able to successfully train our network even without data augmentation, and we can therefore perform a direction comparison.

We apply our method, a variant without data augmentation, and the VQVQE method on three publicly available datasets: "Nancy" from Blizzard 2011 (King & Karaiskos, 2011), Blizzard 2013 (King & Karaiskos, 2013) and LJ (Ito, 2017) dataset. The generated samples are obtained by converting an audio produced by the Google Cloud TTS robot to these three voices. The models are evaluated by their quality using the Mean Opinion Score, as obtained with the CrowdMOS (Ribeiro et al., 2011) package.

As can be seen in Tab. 2, samples generated by our WaveNet autoencoder based method are of higher quality than those of VQ-VAE. A second results is that the method trains well in voice conversion, even without the data augmentation. However, this leads to inferior results.

### A.1  VOICE CONVERSION ARCHITECTURES

We slightly modify the WaveNet autoencoder used in our method for the voice conversion task. Specifically, we modify the size of the latent encoding to be in $\mathbb{R}^{48}$, instead of $\mathbb{R}^{64}$. The rest of the model details remain the same as in the music translation task.

In our implementation of the VQ-VAE, the encoder was composed of 6 one-dimensional convolution layer with a ReLU activation. As in the original paper, the convolutions were with a stride of 2 and

Table 2: MOS scores (mean± SD) for the unseen speaker conversion.

|  | Blizzard 2013 | Nancy | LJ |
|---|---|---|---|
| Our method | $3.16 \pm 0.79$ | $3.85 \pm 0.84$ | $3.40 \pm 0.77$ |
| Our method - without augmentation | $3.07 \pm 0.79$ | $3.87 \pm 0.85$ | $2.85 \pm 0.92$ |
| VQ-VAE | $2.53 \pm 1.08$ | $2.92 \pm 0.92$ | $2.22 \pm 0.96$ |

kernel size of 4. Therefore, the mu-law quantized waveform is temporally downsampled by $\times 64$. We used a dictionary of 512 vectors in $\mathbb{R}^{128}$. The obtained quantized encoding is upsampled and serves to condition a decoder which reconstructs the input waveform. Here as well, we follow the original paper and implement a single WaveNet decoder for all three speaker domains, this is achieved by concatenating the quantized encoding with a learned speaker embedding. We train the VQ-VAE using dictionary updates with Exponential Moving Averages (EMA) with a decay parameter of $\gamma = 0.99$ and a commitment parameter of $\beta = 1$.

