# OpenReview forum: "A Universal Music Translation Network"
_ICLR.cc/2019/Conference_

### Official Review · AnonReviewer2 · 2018-11-01
**review of the paper**

**Rating:** 6
**Confidence:** 4

**Review:**

This paper talks about music translation using a WaveNet-based autoencoder architecture.  The models are trained on diverse training sets and evaluated under multiple settings.  What reported in this paper seems to be interesting and the performance sounds good. However, I have following comments/concerns.

1. The paper is not clearly written. Its exposition needs significant improvement.  There are numerous inconsistent definitions and vague descriptions that make the reading sort of difficult.
    a)  It would be very helpful if the authors can put up a figure for the description of  the WaveNet  autoencoder instead of just using words in Section 3.1
    b) The paper itself should be self-contained instead of referring readers to other references for the details of model architectures.
    c) The math symbols are poorly defined.  What is the definition of C in Section 3.3?   It is defined or referred to as "domain classification network" and also "domain confusion network" but nowhere to find in Fig. 1.
   d) "C is minimizes" -> "minimizes"
   e)  In Section 4,  it says that "Each batch is first used to train the adversarial discriminator".  Which adversarial discriminator? Where to find in Fig. 1 as it is the only description of the network architecture?

2.  The authors mentioned a couple of observations that left unanswered.
    a)   I am surprised to see that without data augmentation, the training does not even converge.
    b)  The conversion from unseen domains is more successful than the learned domains.
    c)  The decoder starts to be creative when the size of the latent space is reduced.
   I sense that these observations seem to point to some (serious) generalization issues of the proposed model.  I would like to hear explanations from the authors.


After reading the rebuttal:
The authors have addressed my major concerns with regard to this paper.   I have lifted my score.  Thanks for the nice response.

---

> ### Author Response · Authors · 2018-11-18
> **Thank you for your constructive comments**
>
> We are sorry for the inconsistencies that were identified in our writing. In the revised version we made the necessary corrections and address all requests for elucidation.
>
> Referring to the issues marked by (a)--(e) in the review:
> (a) We added a figure depicting the architecture of the wavenet autoencoder.
> (b) We added architectural details on the wavenet autoencoder.
> (c) We now term the network C “the domain classification network” and the loss term “the domain confusion term”.
> (d) Fixed.
> (e) By the adversarial discriminator, we meant the network C. The figure is updated.
>
> The review points to three generalization-related phenomena that were mentioned in the paper, which seem unintuitive to the reviewer who asks for clarifications.
>
> (a) Training without data augmentation: While the idea of a denoising autoencoder is not new, this might be the first time in which it is demonstrated that without the added noise, training would fail. To show that this is indeed the case and not an issue with the experiments, we have trained with and without this term for the problem of voice conversion. As noted by R1, this problem is easier than music conversion, and it is, therefore, a convenient testbed.
>
> The details are provided in the new Appendix A, where we also report the MOS score of each model, as obtained with the CrowdMOS package. As can be seen, our method - trained with data augmentation, is better at converting a previously unseen speaker.
>
> In music, we verified again that training without augmentation in unstable. Specifically, without it, we were unable to balance between the domain classification network C and the encoder E, since E is very strongly inclined to memorize the data.
>
> (b) Unseen domains translated better than seen ones: While the training domains used are all classical music, the out of domain examples are more exotic (Swing Jazz, metal guitar riffs, and instrumental Chinese music). Translation, in this case, is often less expected and seems to be more impressive to humans. We hypothesize that this is the reason why under this scenario human raters tend to provide higher translation-success MOS for our output.
>
> (c) Bottleneck effect on the latent space: As requested by the reviewer, we performed an ablation study on the latent code size. Samples are available as a new supplementary - S6, in which we convert a simple MIDI clip to Beethoven domain and Bach domain.
>
> As can be heard, a latent dimensionality of 64 tends to reconstruct the input (unwanted memorization). A model with a latent space of 8 (used in S2) performs well.  A model with a latent dimensionality of 4 is more creative, less related to the input midi, and also suffers from a reduction in quality.

---

> > ### Comment · AnonReviewer2 · 2018-11-28
> > **thanks for the response**
> >
> > Thanks to the authors for the response which is nicely written with some follow-up experiments and explanation to clear my major concerns regarding this paper.   I am willing to raise my score.

---

### Official Review · AnonReviewer1 · 2018-11-04

**Rating:** 7
**Confidence:** 4

**Review:**


The paper proposes a multi-domain music translation method. The model presents a Wavenet auto-encoder setting with a single (domain independent) encoder and multiple (domain specific) decoders.  From the model perspective, the paper builds up on several exciting ideas such as Wavenet and autoencoder based translation models that can perform the domain conversion without relying on parallel datasets. The two main modifications are the use of data augmentation, the use of multiple decoders (rather a single decoder conditioned on the output domain identity) and the use of a domain confusion loss to prevent the latent space to encode domain specific information. This last idea has been also used on prior work.

Up to my knowledge, this is the first autoencoder-based music translation method. While this problem is very similar to that of speaker conversion, modeling musical audio signal (with many instruments) is clearly more challenging.

Summarizing, I think that the contributions in terms of methods are limited, but the results are very interesting. The paper gives an affirmative answer to the question of whether existing models could be adapted to handle the case of music translation, which is of value. The paper would be stronger in my view, if stronger baselines would be included. This would show that the technical contributions are better than alternative methods. Please read bellow some further comments and questions.

The authors perform two ablation studies: eliminating data augmentation and the domain confusion network. In both cases, the model without this add on fails to train. However, it seems to me that different studies are important.

The paper seems to be missing baselines. The authors could compare their work with that of VQ-VAE. The authors claim that they could not make VQ-VAE work on this problem. The cited work by Dieleman et al provides some improvements to adapt VQ-VAE to be better suited to the music domain. Did you evaluate also autoregressive discrete autoencoders?

The proposed method uses an individual decoder per domain. This is unlike other conversion methods (such as the speech conversion studied in VQ-VAE). This modification is very costly and provides a very large capacity. Have you tried having a single decoder which is also conditioned on a one-hot vector indicating the domain? Is it reasonable to expect some transfer between domains or are they too different? Maybe this is the motivation behind using many decoders. It would be good to clarify.

I understand that the emphasis of this work is on music translation, however, the model doesn't have anything specific to music. In that regard, maybe a way to compare to VQ-VAE is to run the proposed method to the voice conversion of the VQ-VAE.

Have you tried producing samples using the decoder in an unconditional setting?

The authors claim that the learned representation is disentangled. Why is this the case? Normally a representation is said to be disentangled if different properties are represented in different (disjoint) coordinates. I might not be understanding what is meant here.

The loss used by the authors, encourages the latent representation to not have domain specific information. The authors should cite the work [A], which has very similar motivation. It would be interesting to report the classification accuracy of the classifier to see how much of the domain information is left in the latent codes. Is it reduced to chance?

In Section 3.1 the authors describe some modifications to nv-wavenet. I imagine that this is because it leads to better performance or faster training. It would be good to give some more information. Did you perform ablation studies for these?

In the human lineup experiment (Figure 2 b,c and d). While the listeners fail to select the correct source, many of the domains are never chosen. This could suggest that some translations are consistently poorer than others or the translations themselves are poor. This cannot be deduced from this experiment. Have you evaluated this?  Maybe it would be better to present pairs of audios with reconstruction and a translation.

While I consider the results quite good, I tend to agree with the posted public comment. It is very hard to claim that the model is effectively transferring styles. A perceptual test should include the question: is this piece on this given style? As the authors mentioned, it is clearly very difficult to evaluate generative models. But maybe the claims could be toned down.

[A] Louizos, Christos, et al. "The variational fair autoencoder." arXiv preprint arXiv:1511.00830 (2015).

---

> ### Author Response · Authors · 2018-11-18
> **Thank you for your constructive and detailed comments**
>
> Thank you for your very constructive comments.
>
> In the revised version, we directly compare our method to the VQ-VAE baseline for a related task for voice conversion, which as you noted, is somewhat easier than music conversion and is also where VQ-VAE was applied to. It is, therefore, a convenient test bed when comparing to the VQ-VAE method, which, did not perform well in our music-based experiments. As can be seen in Tab 2 of Appendix A, samples generated by our method are of higher quality than those of VQ-VAE.
>
> The idea of using a single decoder, which is conditioned on the output domain, is very attractive and has not escaped us. In addition to training fewer networks, it would also support performing domain arithmetics, i.e., creating new domains by combining the conditioning parameters. We started with such experiments as soon as the multi-decoder architecture started to produce results and have invested a considerable effort into this.
>
> Unfortunately, our single-decoder network was never shown to be viable. We attempted multiple conditioning methods, gradually adding to the amount of conditioning from the last layers to earlier and earlier layers. In all cases, the network ignored the domain-conditioning parameters. Our hypothesis is that when trained using teacher forcing, the information on the domain is easily obtained from the previous time frames, causing this neglect. We tried to overcome this by introducing additional loss terms but were not yet successful.
>
> Disentanglement: We thank the reviewer for making this point and this is now clarified in the revised version. We meant to say that the representation disentangles (makes independent, detaches) the domain information from the other modeling aspects.
>
> We have added to [A], which indeed addresses a similar motivation. Their solution is based on the Maximum Mean Discrepancy (MMD), which we do not use.
>
> The paper was updated with Fig 4, depicting the discriminator confusion and accuracy. As can be seen in the confusion matrices, the domain classification network does not do considerably better than chance when the networks converge.
>
> Modifications to nv-wavenet: the NVIDIA kernels implement the architecture of BAIDU, which performs very well for speech. However, in our experiments (in PyTorch), it did not perform as well as the NSynth architecture, which we end up using. We did not find a way to fix this without changing the architecture. Since we did not continue training with the BAIDU architecture, we feel uncomfortable to execute a full blown MOS experiment on this. We are releasing our modified kernels in order to provide the additional option to other researchers.
>
> Lineup experiments: The domain for which the MOS is highest among the six domains is the Mozart symphony domain, which is often selected when the origin is falsely identified. However, audio fidelity does not fully explain the preference of one domain over another in our lineup experiments. The next-best domain, Bach organ (#4), achieved very similar MOS to Mozart symphonies (0.06 difference), yet it was chosen less often than Bach cantatas (#3). #3 achieved the second-lowest MOS. We hypothesize that there are many biases at play. One reason might be the relative complexity --- due to the way computer-generated music is perceived, people tend to assume that “simpler” domains are the outcome of the translation. Additionally, when we output domains that contain singing, it is sometimes possible to note that the singing is not comprised of words.
>
> Reviewer: While I consider the results quite good, I tend to agree with the posted public comment. It is very hard to claim that the model is effectively transferring styles. A perceptual test should include the question: is this piece on this given style? As the authors mentioned, it is clearly very difficult to evaluate generative models. But maybe the claims could be toned down.
>
> Authors: Since our training data was from a limited number of styles (especially now that we moved to publicly available data), the question to which of the domains the generated audio belongs to would be answered in perfect accuracy. This, however, should be credited to WaveNet more than to us --- the decoders are able to generate distinguishable music in the given domains.
>
> If the reviewer means that we should ask “does this piece constitute a perfect, artifact-free, sample from this domain”, then this is similar to what we try to answer with the quality MOS experiments.

---

### Official Review · AnonReviewer3 · 2018-11-07
**promising results, well-written**

**Rating:** 8
**Confidence:** 4

**Review:**

A method is presented to modify a music recording so that it sounds like it was performed by a different (set of) instrument(s). This task is referred to as "music translation". To this end, an autoencoder model is constructed, where the decoder is autoregressive (WaveNet-style) and domain-specific, and the encoder is shared across all domains and trained with an adversarial "domain confusion loss". The latter helps the encoder to produce a domain-agnostic intermediate representation of the audio.

Based on the provided samples, the translation is often imperfect: the original timbre often "leaks" into the output. This is most clearly audible when translating piano to strings: the percussive onsets of the piano (due to the hammers hitting the strings) are also present in the translated audio, even though instruments like the violin and the cello are not supposed to produce percussive onsets. This gives the result an unusual sound, which can be interesting from an artistic point of view, but it is undesirable in the context of the original goal of the paper.

Nevertheless, the results are quite impressive and for some combinations of instruments/styles it works surprisingly well. The question of whether the approach is equivalent to pitch estimation followed by rendering with a different instrument is also addressed in the paper, which I appreciate.

The paper is well written and the related work section is comprehensive. The experimental evaluation is thorough and extensive as well (although a few potentially interesting experiments seemingly didn't make the cut, see other comments). I also like that the authors went through the trouble of doing some experiments on a publicly available dataset, to facilitate reproduction and future comparison experiments.


Other comments:

* "autoregressive" should be one word everywhere

* In section 2 it is stated that attempts to use a unified decoder with style/instrument conditioning all failed. I'm curious about what was tried specifically, it would be nice to discuss this.

* The same goes for experiments based on VQ-VAE, the paper simply states that they were not able to get this working, but not what experiments were run to come to this conclusion.

* The authors went through the trouble to modify the nv-wavenet inference kernels to support their modified architecture, which I appreciate -- will the modified kernels be made available as well?

* The audio augmentation by pitch shifting is a surprising ingredient (but according to the authors it is also crucial). Some more insight as to why this is so important (rather than simply stating that it is important) would be a welcome addition.

* Section 3.2: "out off tune" should read "out of tune".

* The formulation on p.7, 2nd paragraph is a bit confusing: "AMT freelancers tended to choose the same domain as the source, regardless of the real source and the presentation order." Does that mean they got it right every time? I suspect that is not what it means, but that is how I read it initially.

* I don't quite understand the point of the semantic blending experiments. As a baseline, the same kind of blending in the raw audio space should be done, I suspect it would probably be hard to hear the difference. This is how cross-fading is already done in practice, and it isn't clear to me why this method would yield better results in that respect. The paper is strong enough without them so these could probably be left out.

---

> ### Author Response · Authors · 2018-11-18
> **Thank you for your constructive comments**
>
> We thank the reviewer for the thorough review and kind comments.
>
> We have corrected the writing inconsistency of autoregressive and the “out off tune” typo.
>
> Since a single-decoder, in addition to training fewer networks, would also support the creation of new domains by changing the domain-specific parameters, we invested quite a lot in this direction. Unfortunately, our single-decoder network was never shown to be viable. We attempted multiple conditioning methods, gradually adding to the amount of conditioning from the last layers to earlier and earlier layers. In all cases, the network ignored the domain-conditioning parameters. Our hypothesis is that when trained using teacher forcing, the information about the domain is easily obtained from the previous time frames, causing the observed neglect of the domain conditioning. We tried to overcome this by introducing additional loss terms but were not yet successful.
>
> In the revised version, we directly compare our method to the VQ-VAE baseline for a related task for voice conversion. As noted by R1, voice conversion, which is where VQ-VAE was applied before, is not as challenging as music conversion of the type we perform. It is, therefore, a convenient test bed when comparing to the VQ-VAE method, which did not perform well in our music-based experiments.
> The reviewer also mentioned training without augmentation, which, as reported, failed in music. Here, too, voice conversion is a good testbed since it is somewhat easier. We were able to successfully train our network on this task with and without data augmentation.
> Both voice conversion experiments are reported in Appendix A of the revised manuscript. The advantage of using pitch augmentation is clear, and so is the performance gain in comparison to VQ-VAE.
>
> Our entire code, including the modified nv-wavenet inference kernels, will be released in full.
>
> Amazon turkers were biased to always pick the same domain as the original source. We have made this clearer in the revision.
>
> Blending Experiments: We updated our blending experiments following your review. Following the comment, we added to S5 blending done in the WAV domain. Indeed, as you anticipated, for music conversion the differences are small. Still, latent blending probably adds the ability to blend inputs from two different domains (untested).
>
> In order to emphasize the difference between wav-domain and latent-domain blending, we added blending experiments on the voice conversion task. The blended voice samples show a clear difference between the two. Samples blended in WAV space depicts a “cross-fading” effect i.e. a dominant speaker and a quite speaker are heard simultaneously. In contrast, blending in the latent space creates the effect of natural-sounding mumbling of a single speaker.

---

> > ### Comment · AnonReviewer3 · 2018-11-21
> > **reviewer's response**
> >
> > I'm happy to see the additional experiments on voice conversion, and I think it is commendable that the authors went through the trouble of adding these. They sometimes demonstrate some of the claims made in the paper more clearly, so this is a valuable addition (as is the comparison with a VQ-VAE baseline). I'm also happy to hear that the code will be released in full!
> >
> > I hope the other reviewers will take these extensive additions into account and consider revising their reviews accordingly.

---

### Public Comment · ~Keunwoo_Choi1 · 2018-10-02
**Few questions**

Thanks for a nice work. It seems an obvious improvement for me. I also appreciate the detailed subjective evaluation. I have several questions and comments as below.

1. In Section 3.2, what does "and modulate its pitch by a random number between -0.5 and 0.5 of half-steps" mean precisely?

2. In Section 4. Experiments, I would like to hear why all the domains are classical music. Most of each domain consists of single instrument, which would be nice to be mentioned in the paper because - it is probably easier to learn certain instrument than learn the styles of composers, which is better to be elaborated.

3. In Section 4.1 Lineup experiment  -- "we evaluate the ability of persons to identify the source musical segment from the conversions." and also NSynth pitch experiments → What are the goals of these evaluations? What'd be good/bad? What are expected/hoped?

4. In Section 4.2 Are we doing more than timbral transfer? -- "or can it capture stylistic musical elements" → Unlike the former phrase, ("pitch estimation followed by rendering..."), this is quite unclear; 'more than timbral transfer' can mean a wide variety of change; therefore a clear description can be helpful here.
An apparent 'style' transfer that is beyond timbral transfer would be some changes over time, e.g. rhythmic change. However, the observed cases mentioned in the same paragraph sound like they're still limited within some changes along frequency axis, which is not too wrong to be (somewhat roughly) said as a timbral transfer (which is still not bad at all).

Actually, on regarding "There are many samples where it is clear that more than timbral transfer is happening.", there are also samples where the network is merely achieving the "pitch estimation followed by rendering...". For example, in S1 - Opera-to-Bach_solo_piano, the network fails to capture the onsets of notes (there are much more notes in the transferred example). More importantly, it does not sound like "Bach piano", it only sounds like "piano". I think this is a major difference and should be mentioned somewhere in the paper.

5. In Section 4.2 Universality -- Without further analysis, I don't think we can call the network has universality in transferring from unseen instrument (or sound e.g. clapping); that we have some 'result' from unseen instrument doesn't make it universal. It would be very hard to define what is desired in the output when input is something unseen (and clapping could be an extreme case which we might even not need to consider); so how could we interpret it as a proof of universality? The experiment was done with somehow *universal input types*, which is really interesting. But the result doesn't seem to justify any universality of the transfer.

---

> ### Author Response · Authors · 2018-10-04
> **Thank you for posting your questions**
>
> Thank you very much for your detailed comment!
>
> 1. Augmentation code snippet is below.
>
> class WavFrequencyAugmentation:
>     def __init__(self, wav_freq, magnitude=0.5):
>         self.magnitude = magnitude
>         self.wav_freq = wav_freq
>
>     def __call__(self, wav):
>         length = wav.shape[0]
>         perturb_length = random.randint(length // 4, length // 2)
>         perturb_start = random.randint(0, length // 2)
>         perturb_end = perturb_start + perturb_length
>         pitch_perturb = (np.random.rand() - 0.5) * 2 * self.magnitude
>
>         ret = np.concatenate([wav[:perturb_start],
>                               librosa.effects.pitch_shift(wav[perturb_start:perturb_end],
>                                                           self.wav_freq, pitch_perturb),
>                               wav[perturb_end:]])
>
>         return ret
>
> 2. We picked classical music because we feel that it is more straightforward to define and evaluate translating one domain to another within classical music.
>
> We do not focus on single instruments and, in our experience, multi instrument domains do not pose a challenge. Our second network is trained on more single instrument-domains than our previous trained network, and that is because these were the domains found in the open MusicNet.
>
> In particular, our first network was trained on three multi instrument domains out of six total:
> (i) Mozart's symphonies conducted by Karl B"ohm, (ii) Haydn's string quartets, performed by the Amadeus Quartet,  and (iii) J.S Bach's cantatas for orchestra, chorus and soloists. The second network was trained on two (out of six) domains that are multi instrument: (iii) Cambini's  Wind Quintet, (vi) Beethoven's string quartet.
>
> It would be interesting to train on Pop CDs, however it requires proper screening in order to separate to relatively homogeneous styles. It would also be interesting to train on non-classical instrumental music, such as Jazz.
>
> 3. A. human line up experiment -- The goal in this experiment is to evaluate the believability of our translations. With perfect conversions, it would be impossible to identify which is the source material and which is the translated output.
> 3. B NSynth -- The correlation visualization serves to show that our encoder produces semantic embedding vectors when given out-of-training music data as input. A similar text exists in https://arxiv.org/abs/1711.00937 section 4.3 bottom, where an extremely simple phoneme classification scheme is built on top of the latent encodings, and its accuracy is measured. Our visualization shows that musical content is more prominent within the latent encodings than domain-specific content, at least as far as the correlation operator is able to detect.
>
> 4. It is true that our current system does not seem to change note timing. However, this does not mean that it does no more than timbral transfer. Changing only the timbre means keeping the pitches as they are; a system that modifies the pitches, e.g by adding more musical notes, cannot be said to do timbral transfer only. We show many samples where there are more notes in the output than in the input. See sample #3, a cello converted to violin and piano. The cello music is converted to piano, and violin notes are added. In sample #9, piano to cello, the right-hand piano part is discarded, and the remainder is converted to cello. Since our system can be observed to add and remove musical notes, it cannot be said to only be doing timbral transfer.
>
> Of course, there are limits to our abilities to mimic the output domains exactly, particularly in out-of-training-distribution domains such as the opera sample, since there are no singers in MusicNet.
>
> 5. We tried to be very clear in the paper on what we mean by universality, as is also reflected in the comment.  Informally, it means that the network can take any audio input and return an output that is both highly relevant to the input and is in the desired output domain. We demonstrate this ability on extreme inputs.
>
> The question raised in the comment is how to evaluate this ability, which is a major concern in all generative work, not just in music, especially in perceptual domains such as vision, voice, etc. We could answer this using  MOS scores (let us know if needed, we’d be happy to collect), but without access to good baselines to compare against, we are confident enough with the results to simply let the readers decide for themselves.

---

### Author Response · Authors · 2018-11-18
**A revised paper and a summary of the responses to the reviewers**


We thank our colleagues for the detailed comments and for their support. All reviewers seem to agree that the reported results are interesting and performance is good.

VQ-VAE
=======
While we report unsuccessful efforts of employing the VQ-VAE method, both R1 and R3  would like to understand this more. As noted by R1, voice conversion, which is where VQ-VAE was applied before, is not as challenging as music conversion of the type we perform. It is, therefore, a convenient testbed when comparing to the VQ-VAE method, which did not perform well in our music-based experiments, and which was shown by the authors to work on voice conversion.

The experiments in which we compared our method to VQ-VAE were performed on three publicly available datasets - “Nancy” from Blizzard 2011, Blizzard 2013 and LJ dataset. We used on out of domain source samples by converting the Google Cloud TTS robot to these three voices (which effectively creates a TTS pipeline for these voices). The models are evaluated by their quality using the Mean Opinion Score. As can be seen in the table below, samples generated by our method are of higher quality than those of VQ-VAE.

The reviewers were also curious about training without augmentation, which, as reported, failed in music. Here, too, voice conversion is a good testbed since it is somewhat easier. For this task, we were able to successfully train our network with and without data augmentation. As can be seen in the table, our method -- trained with data augmentation, is better at converting a previously unseen speaker.

+----------------------------------------------------+------------------+----------------+---------------+
|                                                                    | Blizzard2013| Nancy         |  LJ              |
+----------------------------------------------------+-------------- ---+----------------+---------------+
| Our method                                             | 3.16+-0.79    | 3.85+-0.84 | 3.40+-0.77 |
| Our method - w/o augmentation        | 3.07+-0.79    | 3.87+-0.85 | 2.85+-0.92 |
| VQ-VAE                                                      | 2.53+-0.95    | 2.92+-0.92 | 2.22+-0.96 |
+----------------------------------------------------+------------------+---------------+----------------+

Additionally, following the reviews, we updated our supplementary (MusicTranslation.github.io) with:
1. An ablation study on the size of the latent code, as requested by R2, is added as S6.
2. Samples generated via Unconditional Generation, as requested by R2, are added as S7.
3. Blending samples for voice and blending in the WAV domain were added to S5, following a comment by R3.

---

### Author Response · Authors · 2019-05-05
**Code Release**

We thank the reviewers for recognizing our contribution.

In order to promote reproducibility, we release our source code and models. Our code includes the modified nv-wavenet inference kernels. The code is available in the following link: https://github.com/facebookresearch/music-translation

The Authors

---

### Meta-Review · Area_Chair1 · 2018-12-13
**interesting problem and promising results**

**Confidence:** 5
**Recommendation:** Accept (Poster)

**Metareview:**

The paper describes a method which, given a music waveform, generates another recording of the same music which should sound as if it was performed by different instruments. The model is an auto-encoder with a WaveNet-like domain-specific decoder and a shared encoder, trained with an adversarial "domain confusion loss". Even though the method is constructed mostly from existing components, the reviewers found the results interesting and convincing, and recommended the paper for acceptance.